# Does Iodine Influence the Metabolism of Glucose?

**DOI:** 10.3390/medicina59020189

**Published:** 2023-01-17

**Authors:** Ioannis Ilias, Charalampos Milionis, Lina Zabuliene, Manfredi Rizzo

**Affiliations:** 1Department of Endocrinology, Diabetes and Metabolism, Elena Venizelou Hospital, GR-11521 Athens, Greece; 2Faculty of Medicine, Vilnius University, M. K. Čiurlionio St. 21/27, LT-03101 Vilnius, Lithuania; 3Department of Health Promotion Sciences, Maternal and Infantile Care, Internal Medicine and Medical Specialties (Promise), School of Medicine, University of Palermo, Via del Vespro, 141, 90127 Palermo, Italy

**Keywords:** iodine, diabetes, thyroid, glucose metabolism, oxidation

## Abstract

Thyroid function and glucose status are linked; experimental, clinical, and epidemiological studies have shown this. Iodine is a vital trace element that is inextricably linked to thyroid hormone synthesis. The latter is also associated with glucose metabolism and diabetes. Recently, some—but not all—studies have shown that iodine is linked to glucose metabolism, glucose intolerance, impaired fasting glucose, prediabetes, diabetes mellitus, or gestational diabetes. In this concise review, we review these studies, focusing on iodine and glucose metabolism and prediabetic conditions or type 2 diabetes mellitus. The potential beneficial effect of iodine on glucose metabolism may be attributed to its antioxidant properties.

## 1. Introduction

Thyroid function and glucose metabolism are linked; this has been shown by experimental, clinical, and epidemiological studies [1]. Iodine (I_2_) is a vital trace element that is inextricably linked to thyroid hormone (THs) synthesis. The latter is also associated with glucose metabolism and diabetes [1]. Recently, studies have shown that I_2_ is linked to glucose metabolism, glucose intolerance, impaired fasting glucose (IFG), prediabetes, and diabetes mellitus (DM) [2,3,4,5]. In this concise review, we focus on I_2_ and glucose metabolism and prediabetic conditions or type 2 diabetes mellitus (DM2).

## 2. Iodine and Thyroid Function/Disease

The incorporation of iodine is a critical step in the biosynthesis of THs. The latter is derived from thyroglobulin (Tg), which is a large dimeric glycoprotein. The thyroid gland extracts up to 10% of iodine from the bloodstream in normal conditions. This process is mediated by the sodium/iodide symporter (NIS) at the basolateral membrane of thyroid follicular cells. NIS is selectively expressed in the thyroid gland, but low levels are also present in the salivary glands, gastric mucosa, kidney, prostate, placenta, lactating breast, and other tissues [6,7]. The uptake of circulating iodine by the thyroid gland is highly adapted to variations in dietary iodine intake. A low supply of iodine stimulates uptake through increased expression of NIS, while high iodine levels have the opposite effect. Within the cytoplasm of thyrocytes, iodine is transported to the apical membrane, where pendrin and other local transporters mediate iodine efflux into the lumen [6,7]. At the extracellular surface of the apical membrane, iodine is oxidized in a reaction that involves thyroid peroxidase (TPO) and hydrogen peroxide (H_2_O_2_). Iodine radicals are added to specific tyrosyl residues within Tg (organification of iodine), thereby generating monoiodotyrosine (MIT) and diiodotyrosine (DIT). The iodotyrosines in Tg are then coupled via an ether linkage with the mediation of TPO. The coupling of two residues of diiodotyrosine (DIT) forms thyroxine (T4), and the coupling of one MIT to one DIT produces triiodothyronine (T3). Then, Tg is transported back into the thyroid cell, where it is processed in lysosomes to release T4 and T3, which are, in turn, secreted into the bloodstream. Uncoupled MIT and DIT can be deiodinated by dehalogenase, a transmembrane enzyme localized mainly at the apical pole of thyrocytes and involved in the intrathyroidal recycling of iodine [6,7].

The World Health Organization (WHO) estimates that about 2 billion people suffer from iodine deficiency. In areas with low levels of iodine, there is an increased prevalence of goiter and hypothyroidism [8]. Iodine deficiency during pregnancy and infancy is still an important cause of worldwide neurological and psychological deficits in children. In adults, mild-to-moderate iodine deficiency may lead to compensatory thyroid enlargement, and hypothyroidism may occur in severe cases. However, excesses in nutritional iodine also have the potential to impact thyroid function. Although most euthyroid individuals can tolerate high iodine intakes, excessive iodine may precipitate hyperthyroidism, hypothyroidism, goiter, and/or thyroid autoimmunity in some people [9]. After exposure to increased iodine levels, thyroid hormone synthesis is inhibited (Wolff-Chaikoff effect). Subjects with mild autoimmune thyroid disease (such as Hashimoto’s thyroiditis) are vulnerable after excessive exposure to iodine, and thyroid dysfunction may fail to resolve after the iodine levels drop in these individuals [9]. Also, in a few people with goiter caused by iodine deficiency, even moderate supplementation with the specific element can lead to autonomous overproduction of THs (Jod-Basedow effect) [9]. Finally, an abrupt elevation in iodine intake may induce thyroid autoimmunity in inhabitants of iodine-deficient areas. Reciprocally, an acute elevation of the iodine intake in subjects with laboratory findings of thyroid autoimmunity (positive antithyroid antibodies) increases their risk of developing thyroid dysfunction [10].

## 3. The Thyroid and Diabetes

Thyroid disease and DM2 are the two most common endocrine disorders treated in clinical practice; associations between them have been reported [11,12,13,14,15]. In the NHANES III study, approximately 14% of all adults had either some form of DM or IFG. In the same study, hypothyroidism was found in 4.6% and hyperthyroidism in 1.3% of the population [16].

Clinical hyperthyroidism has been associated with glucose intolerance [13], whereas hypoglycemia has been reported in patients with hypothyroidism. The pathways involving the participation of THs in the regulation of glucose homeostasis include the induction of hepatic glucose production [17], transcription of mitochondrial genes [18], and expression of genes such as GLUT-4 [19] or phosphoglycerate kinase (PGK) [20].

Several studies have documented a higher rate of thyroid disease in patients with DM compared to individuals without DM: this is highest in up to one-third of women with type 1 diabetes mellitus (DM1) [21]. A threefold to fivefold increase in the risk of autoimmune thyroiditis was observed in patients with positive antibodies to glutamic acid decarboxylase (anti-GAD) [22]. This was confirmed by a study involving 1419 children with DM1, in which 3.5% had Hashimoto’s thyroiditis (HT) [23]; it has to be noted, however, that HT and DM1 may share a common viral causative agent. In addition, positive antibodies against thyroid peroxidase (anti-TPO) have been reported in approximately one-third of patients with DM1 and appear to have prognostic value for the development of clinical and subclinical hypothyroidism [24]. The association between autoimmune thyroiditis and DM1 has been identified as a variant of autoimmune polyglandular syndrome type 3 (APS3) [25,26]. The genetic link between autoimmune thyroiditis and DM1 keeps expanding [27,28]. Given that the prevalence of DM2 is almost 40-fold higher than that of DM1 [29], as indicated in the introduction, in the remainder of this review, we will deal with I_2_ and DM2.

## 4. I_2_ and Glycemia/Diabetes

The prevailing paradigm is that I_2_ exerts its actions via THs (to which it is attached) and, more particularly, via triiodothyronine (T3), which is the biologically active thyroid hormone that binds to thyroid hormone receptors. Indeed, alterations in THs have been associated with glucose metabolism and DM of various degrees of severity (as presented above) [30]. Recently, research works have studied the effect of I_2_ per se on the appearance of Metabolic Syndrome (MetS), including/and/or IFG, impaired glucose tolerance (IGT), prediabetes, or overt DM2 [30]. In most of these studies, I_2_ nutritional adequacy was assessed with the use of urine iodine concentration (UIC), which is considered to be an adequate measure of I_2_ content in the human body [31].

The bulk of the studies on I_2_ and glycemic parameters come from China; this is not surprising, given the scale and the important progress in this country’s program for eradicating I_2_ deficiency [32,33]. In studies conducted in China and the USA, among others, a negative association was shown between UIC and the risk of IFG (Table 1). In a study from China, the relationship between median urine I_2_ (MUI) and the appearance of gestational diabetes mellitus (GDM) was negative.

**Table 1 medicina-59-00189-t001:** Selected studies showing a beneficial effect of I_2_ on glycemic parameters.

Country	Subjects	Main Finding(s)
China [34]	N: 1315 men	FPG > 100 mg/dL was noted in 34% of subjects with UIC < 100 μg/L, in 27.8% of those with UIC: 100–199 μg/L and in 2.6% of subjects with UIC > 200 μg/L (*p* = 0.002)
China [35]	N: 51795 adults	U-shaped curve of UIC vs. IGT Subjects with UIC of 500–799 μg/L showed an OR of 0.753 to 0.838 (95% CI: 0.612–0.939) for IGT against those with lower or higher UIC
United States of America [36]	N: 620 women	With UIC < 100 μg/L vs ≥ 100 μg/L: OR for FPG > 100 mg/dL was 1.73 (95% CI: 1.09–2.72) & OR for HOMA-IR ≥ 2.6 was 0.56 (95% CI: 0.32–0.99)
Kingdom of Saudi Arabia [37]	N: 260 adults	UIC was inversely correlated to FPG and insulin levels (r= −0.40 & −0.16, *p* < 001)
Belgium [38]	N: 471 pregnant women	GDM decreased with increasing placental I_2_ (OR: 0.82, 95% CI: 0.72–0.93, *p* = 0.003)
China [3]	N: 567 adults	Inverse correlation between UIC and risk of DM2 (r: −0.26, *p* < 0.001 and OR: 1.01, 95% CI; 1.00–1.03, *p* = 0.009)
China [4]	N: 144 pregnant women	In women with Ι_2_ excess (MUI > 500μg/L) vs. those with adequate I_2_ (MUI: 150–250 μg/L), the OR for hyperglycemia (FPG > 110 mg/dL) was 0.411 (95% CI: 0.172–0.983)
China [4]	N: 237 breastfeeding women	In women with Ι_2_ excess (MUI > 300μg/L) vs. those with adequate I_2_ (MUI: 100–299 μg/L), the OR for hyperglycemia (FPG > 110 mg/dL) was 0.330 (95% CI: 0.141–0.771)

FPG: fasting plasma glucose; UIC: urine I_2_ concentration; IGT: impaired glucose tolerance—positive oral glucose tolerance test; OR: odds ratio; 95% CI: 95% confidence intervals; HOMA-IR: Homeostatic Model Assessment for Insulin Resistance; DM2: type 2 diabetes mellitus; GDM: gestational diabetes mellitus; MUI: median urine I_2._

In sharp contrast to the above, a study from France (Table 2) found a positive relationship between UIC and the risk of having DM. In contrast, another study from Finland found no association between UIC and the appearance of GDM.

**Table 2 medicina-59-00189-t002:** Selected studies not showing a beneficial effect of I_2_ on glycemic parameters.

Country	Subjects	Main Finding(s)
France [39]	N: 71264 women	The risk for DM * was increased from the third UIC quintile and upwards (HR: 1.20 to 1.28, 95% CI: 1.05–1.53) * defined as FPG ≥ 126 mg/dL, random Glu ≥ 200 mg/dL, A1c > 7% or receiving antidiabetic Rx
Finland [2]	N: 448 women	The authors found no association between UIC and the appearance of GDM

DM: diabetes mellitus; UIC: urine I_2_ concentration; FPG: fasting plasma glucose; Glu: glucose, A1c: glycated hemoglobin A1c; Rx: medication; HR: hazard ratio; 95% CI: 95% confidence intervals; GDM: gestational diabetes mellitus.

Thus, the paradigm that emerges, which is not still unanimous, is that I_2_ has a probable beneficial effect vis-à-vis glucose handling. The relevant research works are limited, according to their authors, by the cross-sectional type of the studies, the possible variability in UIC measurements, and their sample size (in some of them). Tentative explanations regarding I_2_ and glucose handling were not put forth; only inferences were drawn (see below). The effect of THs on the appearance of DM2 cannot be easily supported. The production of THs is a quite resilient process and is usually sustained and stable even in conditions of I_2_ insufficiency [40,41,42]. However, another property of I_2_ may be implicated in averting DM2 [43]. It is known that I_2_ can act as an antioxidant [44]. More in detail, I_2_ can be an antioxidant, depending on its concentration. At minute concentrations, I_2_ can induce a strong anti-oxidant effect [45], although there are reports that at I_2_ excess (ascertained by increased UIC), it can be a pro-oxidant factor [45]. Experimental and clinical studies have shown that supplementation with antioxidants lowers glycemia, insulin resistance, and the risk of DM [46,47,48,49]. Thus, it could be inferred that apart from its indirect effects regarding glucose handling via THs, I_2_ could also exert beneficial direct anti-oxidant effects, with repercussions in glucose metabolism and insulin’s action (Figure 1).

## 5. Discussion

Oxidative stress and inflammation are considered to play a pivotal role in the pathophysiology of DM2. Non-enzymatic glycation of enzymes and other proteins, glucose oxidation, increased lipid peroxidation, impaired glutathione metabolism, and decreased vitamin C levels are mechanisms that can lead to the formation of free radicals. The latter can cause damage to cellular processes and also increase insulin resistance which is a pathogenetic factor for DM2 [50]. Moreover, chronic hyperglycemia is associated with inflammatory processes which are predictive of insulin resistance and DM2 occurrence [51].

Iodine could lower the risk of DM2 via antioxidant and anti-inflammatory effects. Indeed, I_2_ or iodide (I^−^), in particular, can provide protection against free radical attack either via the direct participation of I^−^ as an electron donor in scavenging free radicals or through an indirect action of iodine as a cofactor of peroxidases and as an activator of other antioxidant enzymes. Besides the non-hormonal antioxidant properties of iodine, an adequate intake of this element is certainly necessary for optimal thyroid function, which is also a prerequisite for a well-regulated antioxidant status [45]. Iodine also has an anti-inflammatory action by neutralizing radical oxygen species and suppressing pro-inflammatory messengers, such as tumor necrosis factor-a and interleukin-6) [38], and hence, it could have an additional protective effect against DM2.

Iodine supplementation, mainly through the iodization of salt, is an effective public health policy for the prevention of iodine deficiency in the general population. The recommended daily intake of iodine is 150 μg for adults and persons older than 14 years old, 220 μg for pregnant women, and 290 μg for breastfeeding women [52]. Urinary iodine excretion is higher than 100 μg/L in iodine-sufficient populations. However, it is debatable whether these dosages could have a beneficial effect on the risk of DM2. It appears that iodine acts as an antioxidant in the body only if ingested at concentrations higher than 1 mg per day [53]. Nonetheless, safety concerns arise from these levels about the integrity of thyroid function and the occurrence of possible side effects, including liver damage, kidney dysfunction, headache, conjunctivitis, edema of the salivary glands, fever, and skin reactions. It is also noted that Graves’ thyrotoxicosis is a contraindication for iodine therapy. The existence of Hashimoto’s disease is also a concern, especially for higher doses.

Finally, we cannot ignore another overlooked fundamental trace element, Selenium (Se), which is linked with I_2_ and the production of THs and may have an impact on glucose metabolism [54,55,56,57,58].

## 6. Conclusions

The existing evidence on the intra- and extra-thyroidal role of iodine with regard to the development of DM2 is not solid. Therefore, the aforementioned assumptions need to be verified with concrete studies aiming at investigating these aspects of iodine’s actions. We believe that since the interconnection between iodine, thyroid function, and glucose homeostasis seems plausible, and also given the epidemiological status of iodine deficiency and DM2 prevalence worldwide [59,60], further research is warranted.

## Figures and Tables

**Figure 1 medicina-59-00189-f001:**
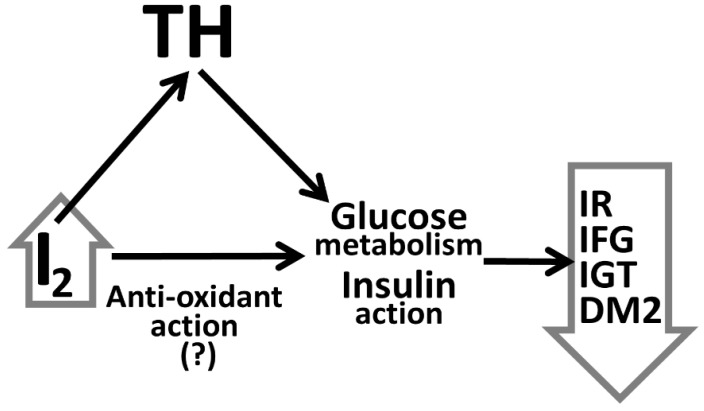
Tentative “triangular” aspect of Iodine’s (I_2_) effects on glucose handling: indirect effects are via the synthesis and the action of thyroid hormones (TH), direct effects could be postulated via anti-oxidant action; IR: insulin resistance; IFG: impaired fasting glucose; IGT: impaired glucose tolerance; DM2: type 2 diabetes mellitus.

## Data Availability

Not Applicable.

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
