# Peer review of "Does Iodine Influence the Metabolism of Glucose?"

_medicina, 2023, doi:10.3390/medicina59020189_

Round 1

Reviewer 1 Report

This paper provides a brief review on interaction of thyroid, iodine and DM. IN this field no firm conclusions can be made, while the data are either sparce or contradictory.

I have two major comments and a few minor.

1. The part 3: "The thyroid and Diabetes" is difficult to read,  Although I do understand that it is not easy to write such an overview, it think it could be rewritten in a more structured und understandable way.

2. In the discussion the last paragraph on Selenium fills about half of the discussion, which is surprising, as it is not the focus of the paper. So I would recommend to shorten this paragraph. The main message should be that Selenium seems to have an important role.

Minor comments.

Line 17 and 28. The verb hone was not familiar to me and I would guess for more in the readership. substitute with "with emphasis on" or "focussed on"?

Line 101 poly instead of poly poly

Author Response

Responses to Reviewer #1

This paper provides a brief review on interaction of thyroid, iodine and DM. IN this field no firm conclusions can be made, while the data are either sparce or contradictory.

I have two major comments and a few minor.

[1]. The part 3: "The thyroid and Diabetes" is difficult to read. Although I do understand that it is not easy to write such an overview, it think it could be rewritten in a more structured und understandable way.

We thank the Reviewer for the comment. The relevant section has been shortened and simplified as follows:

“Thyroid disease and DM2 are the two most common endocrine disorders treated in clinical practice; associations between them have been reported [11-15]. In the NHANES III study, approximately 14% of all adults had either some form of DM or IFG. In the same study, hypothyroidism was found in 4.6% and hyperthyroidism in 1.3% of the population [16].

Clinical hyperthyroidism has been associated with glucose intolerance [13], whereas hypoglycemia has been reported in patients with hypothyroidism. The pathways involving the participation of THs in the regulation of glucose homeostasis include the induction of hepatic glucose production [17], transcription of mitochondrial genes [18], and expression of genes such as GLUT-4 [19] or phosphoglycerate kinase (PGK) [20].

Several studies have documented a higher rate of thyroid disease in patients with DM compared to individuals without DM: this is highest - in up to one third - for women with type 1 diabetes mellitus (DM1) [21]. A threefold to fivefold increase in the risk of autoimmune thyroiditis was observed in patients with positive antibodies to glutamic acid decarboxylase (anti-GAD) [22]. This was confirmed by a study involving 1419 children with DM1, in which 3.5% had Hashimoto's thyroiditis (HT)[23]; it has to be noted, however, that HT and DM1 may share a common viral causative agent. In addition, positive antibodies against thyroid peroxidase (anti-TPO) have been reported in approximately one third of patients with DM1, and appear to have prognostic value for the development of clinical and subclinical hypothyroidism [24]. The association between autoimmune thyroiditis and DM1 has been identified as a variant of autoimmune polyglandular syndrome type 3 (APS3) [25,26]. The genetic link between autoimmune thyroiditis and DM1 keeps expanding [27,28]. Given that the prevalence of DM2 is almost 40-fold higher than that of DM1 [29], as indicated in the introduction, in the remaining of this review we will deal with I2 and DM2. “

[2]. In the discussion the last paragraph on Selenium fills about half of the discussion, which is surprising, as it is not the focus of the paper. So I would recommend to shorten this paragraph. The main message should be that Selenium seems to have an important role.

In the revised version of the manuscript this section has been substantially shortened to a paragraph as follows:

“Finally, we cannot ignore another - overlooked - fundamental trace element, Selenium (Se), which is linked with I2 and the production of THs and may have an impact on glucose metabolism [54-58].”

Minor comments.

[3]. Line 17 and 28. The verb hone was not familiar to me and I would guess for more in the readership. substitute with "with emphasis on" or "focussed on"?

In the revised version of the manuscript "focussed" and "focus on" have substituted "honed" and "hone on".

[4]. Line 101 poly instead of poly poly.

We have corrected this in the revised version of the manuscript.

A final note to the Reviewer: If the Journal desires so, we are providing two revised versions of the manuscript back-to-back: one, which remains a mini-review with tables and a figure, whereas the other is a short communication, per the suggestion of Reviewer #2.

Reviewer 2 Report

The title is misleading and confusing, leading one to think that there is a definite relationship between thyroid hormones and diabetes, however the authors conclude in their conclusion that there is no solid evidence for their assumption. The authors also mix Type 1 and Type 2 diabetes. The association between Hashimoto and TID is probably due to a common viral etiology and not related to iodine.

I suggest the authors write a short communications entitled “Does iodine influence glucose metabolism”

Author Response

Responses to Reviewer #2

[1]. The title is misleading and confusing, leading one to think that there is a definite relationship between thyroid hormones and diabetes, however the authors conclude in their conclusion that there is no solid evidence for their assumption.

We thank the Reviewer for pointing out this paper's conundrum. We fully agree with him/her. To avoid confusion and per point No [4] we have changed the title in the revised version of the manuscript. Please see our response to his/her point No [4].

[2].  The authors also mix Type 1 and Type 2 diabetes.

In the revised version of this manuscript we have changed the phrasing to "some form of DM" in the second sentence of the relevant section and at the end of this section we specify that in the rest of the paper we refer to Diabetes Mellitus type 2.

[3]. The association between Hashimoto and TID is probably due to a common viral etiology and not related to iodine.

In the revised version of this work we mention that Hashimoto's Thyroiditis and DM type 1 may share a common viral causative agent.

[4]. I suggest the authors write a short communications entitled “Does iodine influence glucose metabolism”.

In the revised version of the manuscript the title has been changed to "Does Iodine influence the Metabolism of Glucose?" As for the paper's form/length we have shortened it. Nevertheless, if the Journal desires so, we are providing two revised versions of the manuscript: one, which remains a mini-review with tables and a figure, whereas the other is a short communication, per the Reviewer's suggestion.

Round 2

Reviewer 2 Report

Paper can be accepted